# Farther Than Mirror:
# Explore Pattern-Compensated Depth of Mirror with Temporal Changes for Video Mirror Detection

## Abstract

Current video mirror detection models demonstrate satisfactory performance by analyzing different attributes of mirrors and incorporating temporal information. However, these models still struggle to detect mirrors in complex and dynamic scenarios. A simple yet critical visual cue is that *objects reflected in a mirror appear to be farther away than the mirror itself*. Motivated by this observation, we propose to explicitly analyze the Depth of Mirror (DOM) within a video to effectively localize mirrors - DOM refers to distinct perceived distances that make mirror regions appear farther away from their surroundings. Specifically, we devise a novel framework called FTM-Net, which contains two main contributions: a Pattern-Compensated DOM estimation strategy and a Dual-Granularity Affinity module. The Pattern-Compensated DOM estimation strategy uses multiple visual mirror patterns to refine the DOM, enhancing the accuracy of mirror localization in a single image. Furthermore, the Dual-Granularity Affinity module can effectively detect mirrors in video sequences by tracking and integrating DOM changes across frames. Experimental results on a benchmark dataset show that our model significantly outperforms 18 state-of-the-art methods in the video mirror detection task.

## 1 Introduction

Mirrors are commonly found in our daily lives; however, their presence can have a significant negative impact on various applications, such as drone tracking (Chen et al., 2017b), robot navigation (Gul et al., 2019), and so on. Accurate mirror detection is crucial for avoiding potential safety issues in these applications. Therefore, the development of mirror detection models is essential to precisely detect mirrors and to provide critical mirror information for other tasks. As shown in Figure 1 and according to optical principles (Born & Wolf, 2013; Mei et al., 2021), *objects reflected in a mirror appear to be farther away than the mirror itself.* Motivated by this observation, we introduce a new concept, the Depth of Mirror (DOM), which identifies areas of the image that appear farther away from their surroundings and are likely to be mirrors. We argue that accurately analyzing DOM is crucial for effective mirror detection.

While the existing literature has primarily focused on single-image mirror detection by exploring the attributes of mirrors themselves (Lin et al., 2023; Liu et al., 2023c; Yang et al., 2019), these methods struggle with video mirror detection tasks due to the absence of temporal information across video frames. Recently, VMD-Net (Lin et al., 2023) achieved promising video mirror detection results by utilizing both intra-frame and inter-frame correspondences to model temporal information. However, merely modeling temporal information without considering mirror information proves to be insufficient, as mirrors reflect actual objects, complicating the localization of mirror areas in dynamic scenes. This limitation underscores the necessity for *a comprehensive approach that not only models temporal dynamics but also integrates these with DOM changes* to thoroughly address the challenges of video mirror detection.

To this end, we introduce a novel method named FTM-Net (**F**arther **T**han **M**irror), which features a Pattern-Compensated DOM estimation strategy and a Dual-granularity Affinity module for video

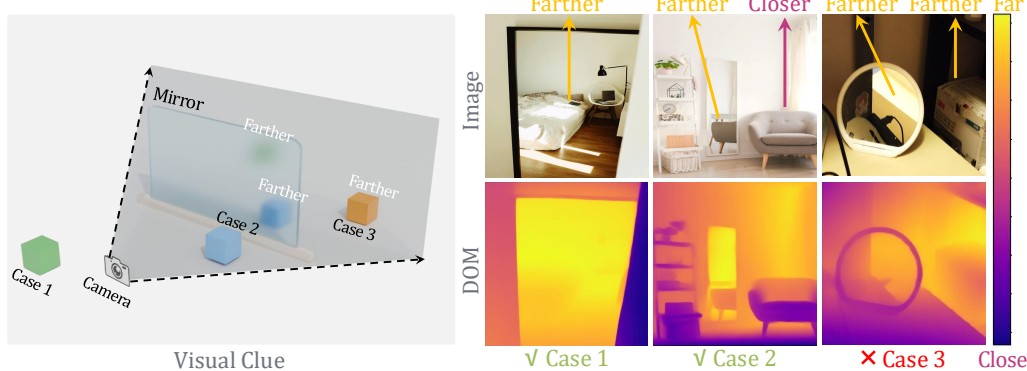

Figure 1: Visual Clue: objects reflected in the mirror appear farther than the mirror itself. Case 1: objects behind the camera. Case 2: objects between the camera and the mirror. Both Case 1& 2's reflections appear farther away than the mirror. Case 3: objects behind the mirror, which also appear far, but must be excluded from the Depth of Mirror (DOM) analysis to ensure accuracy.

mirror detection tasks. The Pattern-Compensated DOM estimation strategy firstly generates a DOM from a depth estimation network to identify regions of the image that are farther away from their surroundings and likely to be mirrors, as shown in Cases 1 & 2 in Figure 1. Considering a scenario illustrated in Case 3 of Figure 1, where non-mirror objects are physically farther than mirrors and might be incorrectly included in the DOM, we further refine and compensate the DOM using multiple mirror patterns. This refinement produces a more precise pattern-compensated DOM. To further enhance the detection and analysis in dynamic mirror interactions, we design a Dual-Granularity Affinity module that integrates both pixel and pattern changes of DOM-related video features into the current frame feature. Our main contributions are summarized as follows:

- We present a novel model named FTM-Net to address video mirror detection tasks, in which we highlight:
  - A novel Pattern-Compensated DOM estimation strategy that integrates DOM with multiple mirror patterns to more accurately detect mirrors in single frames.
  - A Dual-Granularity Affinity module integrates both pixel and pattern changes of video features into the current frame feature, enhancing the temporal representation related to mirror detection.
- Extensive experimental results on a benchmark dataset demonstrate that our FTM-Net outperforms 18 leading state-of-the-art methods for video mirror detection.

## 2 RELATED WORK

**Image Mirror Detection.** Detecting mirrors in an image often works as the first step in various computer vision applications, including drone tracking and robot navigation. Numerous methods have been developed for mirror segmentation tasks. For example, MirrorNet (Yang et al., 2019) introduced the first mirror detection dataset and network, utilizing contextual contrasted information for accurate detection. PMDNet (Lin et al., 2020) presents a more challenging benchmark for mirror detection tasks by leveraging multi-scale mirror edge features to enhance perception. More recently, VCNet (Tan et al., 2022) has further improved mirror detection performance by exploiting chirality cues and implicit correspondences. Mei et al. (Mei et al., 2021) propose the first RGB-D mirror segmentation dataset and utilize depth information to assist in mirror detection. However, this method requires additional devices to simultaneously capture the RGB image and depth map. Furthermore, these methods still fall short when applied to video mirror detection, as they do not consider the temporal relationships between adjacent frames.

**Video Shadow/Saliency/Mirror Detection.** Many video-based methods have been proposed to explore temporal information. Video shadow detection task has been extensively studied in recent years. For instance, Scotch-Soda (Liu et al., 2023c) presents a new type of trajectory attention and employs a contrastive loss to help the model learn more robust representations of shadows. Additionally, Li et al. (Li et al., 2019) propose a motion-guided attention mechanism by incorporating optical flow to enhance appearance features for video saliency detection. Despite these advancements

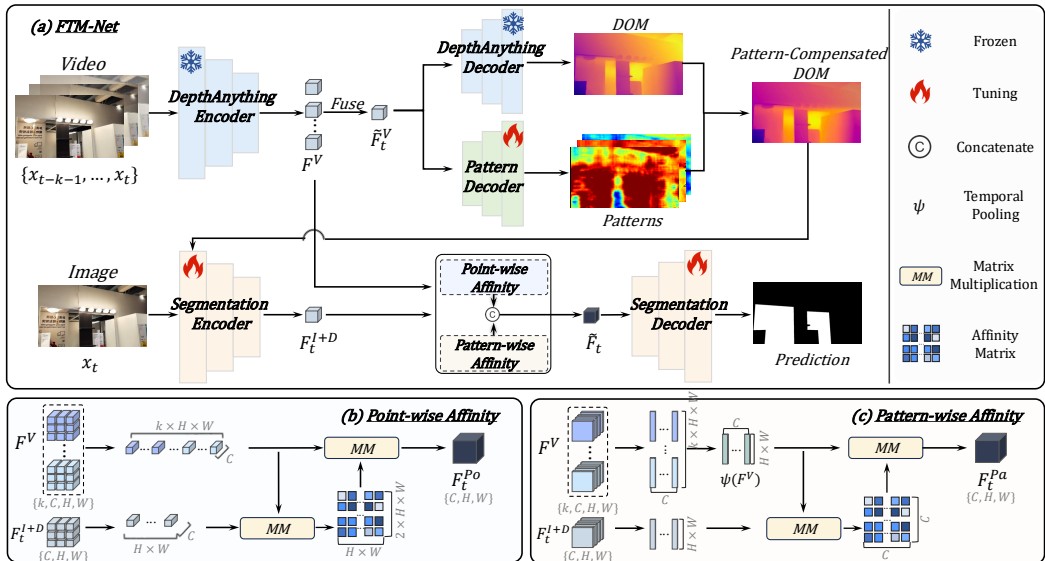

Figure 2: The overview of our FTM-Net. First, the video input is fed into the DepthAnything encoder to extract DOM video features. Then, the DepthAnything and Mirror Pattern decoders are employed to generate a pattern-compensated DOM. The pattern-compensated DOM is integrated into a segmentation encoder with the input frame $x_t$ and the Dual-Granularity Affinity module is used to merge the combined feature $F_t^{I+D}$ with the DOM video feature. Finally, the fused feature $\tilde{F}^I$ is processed through a segmentation decoder to generate the final mirror detection map.

in video shadow/saliency detection, the challenge of video mirror detection has only recently begun to be addressed. VMD-Net (Lin et al., 2023) introduces the first video mirror detection dataset and utilizes both intra-frame and inter-frame correspondences to capture temporal information. However, while its performance is promising, relying solely on temporal information proves insufficient due to the complex reflective attributes of mirrors, which complicate the accurate localization and identification of mirrored surfaces in dynamic scenes.

**Depth Estimation Networks.** Depth estimation is crucial in computer vision for the perception and understanding of real scenes. Eigen et al.(Eigen et al., 2014) introduced the first multi-scale fusion network based on deep learning to predict depth maps. Subsequently, numerous studies have enhanced depth estimation accuracy by incorporating additional priors(Li et al., 2015; Shao et al., 2023; Liu et al., 2023b) or optimizing objective functions (Yin et al., 2019; Xian et al., 2020; Liu et al., 2023a). However, these methods are often limited by data scalability and struggle to generalize well to unseen domains. To address these challenges, recent innovations inspired by the Segment Anything model (SAM) have emerged. Notably, DepthAnything (Yang et al., 2024) designed a foundation model that can generate accurate depth annotations for images across various scenarios in a zero-shot manner.

**Affinity Mechanism for Video Processing.** The affinity mechanism has been proven to be an efficient and effective way to capture video features by modeling the relationship between the features of the current frame and contextual frames, where contextual frames refer to those previous to the current one (Cheng et al., 2021b). In this mechanism, the negative squared Euclidean distance is employed as the similarity function to capture the relationship between features:

$$\mathbf{S}_{i,j} = -\left\|\mathbf{k}_i^M - \mathbf{k}_j^Q\right\|_2^2 = 2 \cdot \mathbf{k}_i^M \cdot \mathbf{k}_j^Q - \left\|\mathbf{k}_i^M\right\|_2^2 - \left\|\mathbf{k}_j^Q\right\|_2^2, \tag{1}$$

where $\mathbf{k}_i^M$ represents the memory key of the current frame and $\mathbf{k}_j^Q$ is the query key of the contextual frames, and the latter expression offers a more efficient way to compute the similarity matrix. Although useful, this affinity module computes only the pixel relationship, overlooking the unique visual patterns of objects represented in the features level (Chu et al., 2016; Wu et al., 2023). To address this limitation, we design a Dual-Granularity Affinity module, which consists of both point-wise and pattern-wise affinity mechanisms. This design improves the temporal representation of objects, such as mirrors or shadows, by better capturing the complex patterns that distinguish them.

## 3 METHOD

In this section, we first introduce a novel Pattern-Compensated Depth of Mirror (DOM) estimation strategy. Next, we provide a detailed explanation of our novel Dual-Granularity Affinity module. Finally, we describe the overall workflow of our FTM-Net.

### 3.1 PATTERN-COMPENSATED DOM ESTIMATION

**DepthAnything for DOM Estimation.** To capture the DOM, which identifies areas of the image that are far from the viewer and likely to contain mirrors, we use DepthAnything as the DOM estimation network. DepthAnything, including an encoder and a decoder, is used without any training process. The parameters of DepthAnything are frozen and used to generate the DOM in a zero-shot manner.

**Pattern Decoder for Enhanced DOM.** Since DepthAnything is used with a frozen state during the training stage, the output DOM may inadvertently include some non-mirror regions that are also physically farther than the mirror itself; as shown in Case 3 of Figure 1. Therefore, relying solely on DepthAnything to estimate DOM is inadequate for accurately detecting mirrors. To overcome this limitation, we introduce a mirror-pattern decoder that outputs multiple potential patterns specifically for mirrors. This decoder shares a similar architecture with the DepthAnything decoder but differs in two key aspects. First, instead of outputting a single channel as the DepthAnything decoder, the mirror-pattern decoder outputs multiple channels, each representing a different mirror pattern. Second, unlike the frozen state of the DepthAnything model, this decoder is actively trained to learn these mirror patterns. The output mirror patterns are then concatenated with the initial DOM map to produce a refined DOM, enhancing mirror detection accuracy.

### 3.2 DUAL-GRANULARITY AFFINITY (DGA)

The affinity mechanism is widely used to extract temporal features by computing relationships between pixels in the current frame and contextual frames, where contextual frames refer to those previous to the current one. By providing a temporal context, this mechanism has proven to be powerful in understanding motion and changes over time. However, this method often overlooks the unique visual patterns of objects (Wu et al., 2023; Chu et al., 2016). To address this limitation, we have designed a Dual-Granularity Affinity (DGA) module, which includes both point-wise and pattern-wise affinity. This module enhances the representation of temporal relationships by capturing not only pixel relationships but also the broader visual patterns that define object behavior in video sequences, as depicted in Figure 2.

**Point-wise Affinity.** The point-wise affinity begins with two features: the current frame Image and DOM combined feature $F_t^{I+D}$ and the Video-level DOM-related feature $F^V$. The point-wise similarity matrix is then calculated as follows:

$$S_{i,j}^{Po} = 2 \cdot (F^V)_i^T \cdot (F_t^{I+D})_j - \left\| (F^V)_i^T \right\|_2^2 - \left\| (F_t^{I+D})_j \right\|_2^2, \tag{2}$$

where $F^V \in \mathbb{R}^{C \times \frac{k \times H \times W}{16 \times 16}}$ and $F_t^{I+D} \in \mathbb{R}^{C \times \frac{H \times W}{16 \times 16}}$. $T$ denotes matrix transpose operator. After computing the point-wise similarity matrix $S_{i,j}^{Po}$, the normalized affinity matrix $\mathcal{W}^{Po}$ is derived. Then, the final output feature $F^{Po}$, which incorporates contextual information from previous frames, is defined as follows:

$$F_t^{Po} = F^V \cdot \mathcal{W}^{Po}, \text{ where } \mathcal{W}^{Po} = \frac{exp\left(S_{i,j}^{Po}\right)}{\sum_x exp\left(S_{x,j}^{Po}\right)} \in \mathbb{R}^{\frac{k \times H \times W}{16 \times 16} \times \frac{H \times W}{16 \times 16}}. \tag{3}$$

**Pattern-wise Affinity.** Pattern-wise affinity also begins with two features: the current frame Image and DOM conbined feature $F_t^{I+D}$ and the video-level DOM-related feature $F^V$. Instead of capturing pixel relationships, this approach aims to capture the relationship between the object in the current frame and the different visual patterns of objects in previous frames. Hence, a temporal pooling operator $\psi$ is employed to maintain spatial dimensions. The pattern-wise similarity matrix $S_{pa}$ is defined as follows:

$$S_{i,j}^{Pa} = 2 \cdot \psi(F^V)_i \cdot (F_t^{I+D})_j^T - \left\| \psi(F^V)_i \right\|_2^2 - \left\| (F_t^{I+D})_j^T \right\|_2^2. \tag{4}$$

After computing the pattern-wise similarity matrix $S_{i,j}^{Pa}$, the normalized similarity matrix $\mathcal{W}^{Pa}$ is derived. Subsequently, the final output for the pattern-wise integrated feature $F^{Pa}$, which incorporates contextual pattern information from previous frames, is defined as follows:

$$F_t^{Pa} = \psi(F^V)^T \cdot \mathcal{W}^{Pa}, \text{ where } \mathcal{W}^{Pa} = \frac{\exp\left(S_{i,j}^{Pa}\right)}{\sum_x \exp\left(S_{x,j}^{Pa}\right)} \in \mathbb{R}^{C \times C}. \tag{5}$$

The final output of our DGA module $\tilde{F}_t$ is derived from the concatenation of $F_t^{Po}$ and $F_t^{Pa}$.

## 3.3 OVERALL WORKFLOW

Figure 2 demonstrates the overall workflow of the FTM-Net. Overall, our method takes one current frame $x_t$, and its $k-1$ before video sequence $V : \{x_{t-k-1}, ..., x_{t-1}, x_t\} \in \mathbb{R}^{k \times 3 \times H \times W}$ as inputs; and output a mirror detection result. Our two inputs are mainly used in two stages: (i) The DOM estimation stage and (ii) the DOM temporal information integration stage.

**DOM Estimation Stage.** In the DOM estimation stage, the video input $V$ is fed into the DepthAnything encoder to extract DOM video features from different frames, denoted as $F^V = \{F_{t-k-1}^V, ..., F_{t-1}^V, F_t^V\} \in \mathbb{R}^{k \times C \times \frac{H}{16} \times \frac{W}{16}}$ , where $C$ is feature dimension. We further fuse the video feature $F^V$ using our DGA module to get an image-level feature $\tilde{F}_t^V$. Then, two decoders (DepthAnything Decoder and Mirror Pattern Decoder) are employed to predict the DOM and mirror patterns, respectively. The outputs DOM and mirror patterns are concatenated to generate a pattern-compensated DOM.

**DOM Temporal Change Integration Stage.** The second stage utilizes three inputs: the current frame $x_t$, DOM video features $F^V$ and the pattern-compensated DOM from the first stage. First, the pattern-compensated DOM is integrated into a segmentation encoder via a simple patch embedding layer with the input frame $x_t$ to generate the DOM-aware image feature $F_t^{I+D}$, where $I$ denotes Image and $D$ represents DOM. Then, we utilize the Dual-Granularity Affinity module to merge the image and DOM combined feature $F_t^{I+D}$ with the DOM-related video feature $F^V$, aiming to achieve a better perception of mirrors. Finally, the fused feature $\tilde{F}^I$ is processed through a segmentation decoder to generate the final mirror detection map for the current frame $x_t$.

## 4 EXPERIMENT

### 4.1 EXPERIMENTAL SETTINGS

**Dataset.** We evaluate our method on the VMD-D dataset, which is the first large-scale video mirror detection dataset consisting of 269 videos in 14,988 image frames with corresponding precise annotations from diverse scenes. We follow the same data splitting setting of (Lin et al., 2023) to divide the entire VMD-D dataset into a training set with 143 videos (7835 images) and a testing set with 126 videos (7152 images). The frame rate is 30 FPS for all videos and the images in each video are with a high resolution of $1920 \times 1080$.

**Evaluation Metrics.** Following previous works (Lin et al., 2023; Tan et al., 2022; Lin et al., 2020), we adopt intersection over union (IoU), pixel accuracy (Accuracy), F-measure ($F_\beta$), and mean absolute error (MAE) to evaluate our method.

**Implementation Details.** Our model is implemented in PyTorch 2.0.1-cuda11.7 and trained on four NVIDIA 4090 GPUs (24G memory for each one) with a batch size of 8. The segmentation model used in our method is SegFormer (Xie et al., 2021), which is initialized using the weights from the Mit-B2 model pre-trained on the ADE20K dataset (Zhou et al., 2017; 2019). We use pre-trained DepthAnything-S (Yang et al., 2024) as our depth estimation network. The remaining parameters are initialized using the Xavier (Glorot & Bengio, 2010) method. During training, we resize all video frames to $512 \times 512$ and use a random horizontal flip for data augmentation. We use an Adam (Loshchilov & Hutter, 2017) optimizer along with a poly learning rate scheduler (an initial learning rate of 1e-3 and a weight decay of 3e-5) and run a total of 15 epochs for all experiments and ablation studies. For inference, we do not apply any post-processing techniques and only resize the resolution of input frames to $512 \times 512$.

| METHODS | | | EVALUATION METRICS | | | |
|---|---|---|---|---|---|---|
| Tasks | Techniques | Type | IoU ↑ | Accuracy ↑ | $F_\beta$ ↑ | MAE ↓ |
| SOD | GateNet (Zhao et al., 2020) | Image | 0.429 | 0.851 | 0.665 | 0.153 |
| | MINet (Pang et al., 2020) | Image | 0.412 | 0.854 | 0.676 | 0.148 |
| IOS | DeepLabV3 (Chen et al., 2017a) | Image | 0.481 | 0.846 | 0.681 | 0.157 |
| | PSPNet (Zhao et al., 2017) | Image | 0.464 | 0.850 | 0.665 | 0.152 |
| | OCRNet (Yuan et al., 2020) | Image | 0.394 | 0.786 | 0.640 | 0.175 |
| | Mask2Former (Cheng et al., 2021a) | Image | 0.547 | 0.862 | 0.691 | 0.137 |
| VSD | TVSD (Chen et al., 2021) | Video | 0.480 | 0.875 | 0.746 | 0.125 |
| | STICT (Lu et al., 2022) | Video | 0.164 | 0.809 | 0.530 | 0.198 |
| | Sc-Cor (Ding et al., 2022) | Video | 0.512 | 0.863 | 0.696 | 0.137 |
| | Scotch-Soda (Liu et al., 2023c) | Video | 0.587 | 0.878 | 0.749 | 0.121 |
| VOS | HFAN (Pei et al., 2022) | Video | 0.459 | 0.876 | 0.706 | 0.124 |
| | STCN (Cheng et al., 2021b) | Video | 0.445 | 0.859 | 0.670 | 0.140 |
| IMD | GlassNet (Lin et al., 2021) | Image | 0.552 | 0.864 | 0.718 | 0.137 |
| | MirrorNet (Yang et al., 2019) | Image | 0.505 | 0.855 | 0.681 | 0.145 |
| | PMDNet (Lin et al., 2020) | Image | 0.532 | 0.872 | 0.749 | 0.128 |
| | VCNet (Tan et al., 2022) | Image | 0.539 | 0.877 | 0.749 | 0.123 |
| | HetNet (He et al., 2023) | Image | 0.531 | 0.868 | 0.748 | 0.131 |
| VMD | VMD-Net (Lin et al., 2023) | Video | 0.567 | 0.895 | 0.787 | 0.105 |
| | Ours | Video | **0.649** | **0.913** | **0.833** | **0.083** |

Table 1: Quantitative comparison between the proposed FTM-Net and 18 state-of-the-art methods from relevant fields on the VMD-D dataset. The ↑ denotes the higher the value is the better the performance is, whilst the ↓ means the opposite.

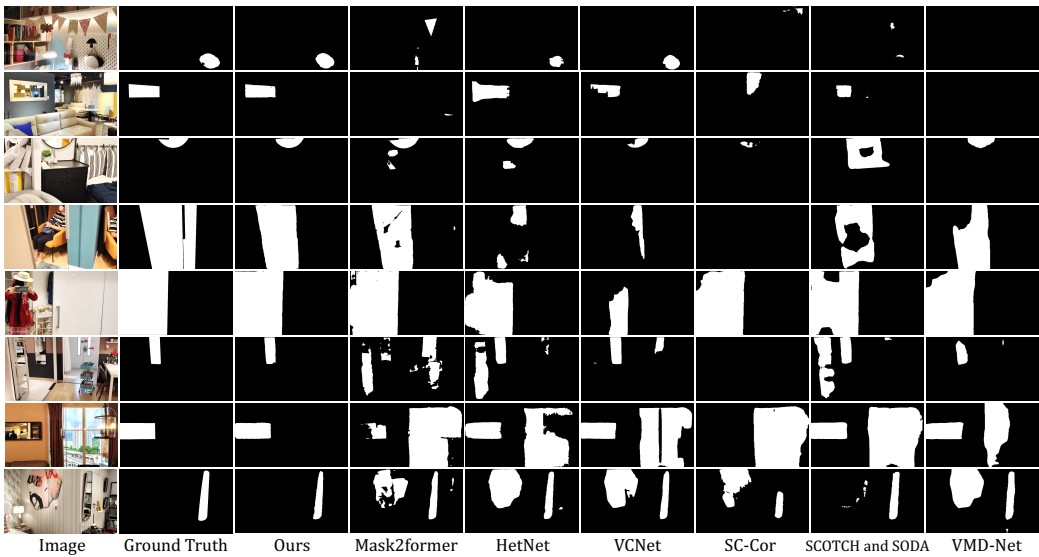

Figure 3: Visual comparisons of video mirror detection results predicted by our FTM-Net and compared methods. Apparently, our network can obtain more accurate mirror detection results than all compared methods. and our results are more consistent with the ground truths. Video results can be found in *supplementary material*.

## 4.2 COMPARISON AGAINST STATE-OF-THE-ART METHODS

**Compared Methods.** Following the same setting of the recent VMD-Net, we first compare our network with 14 state-of-the-art methods, including GateNet and MINet for salient object detection; DeepLabV3, PSPNet, and OCRNet for semantic segmentation; TVSD, STICT, Sc-Cor, and Scotch-Soda for video shadow detection; HFAN for video object segmentation; GlassNet for glass surface detection; as well as MirrorNet, PMDNet, and VCNet for single-image mirror detection. Moreover, we add four new methods for comparisons: Mask2Former for semantic segmentation, Scotch-Soda

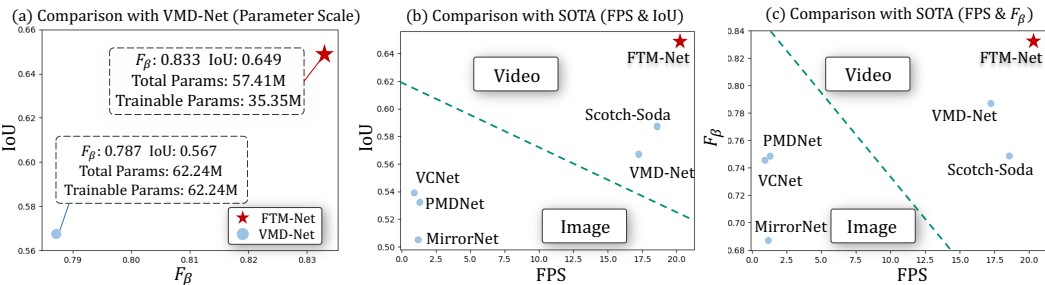

Figure 4: Parameter scale and inference efficiency comparisons. FTM-Net achieves the best performance with fewer parameters.

for video shadow detection, HetNet for single-image mirror detection, and STCN for video object segmentation. We obtain the results on VMD-D dataset by downloading the results from VMD-Net official repository and re-train other methods with unified training parameters to keep fairness.

**Quantitative Comparisons.** As shown in Table 1, Scotch-Soda has the best IoU performance of 0.587, while VMD-Net has the best Accuracy performance of 0.895, the best $F_\beta$ performance of 0.787, and the best MAE score of 0.105 among the 18 compared methods. Compared to VMD-Net, our network outperforms VMD-Net in terms of all four metrics. Specifically, our network improves the IoU score from 0.567 to 0.649, the Accuracy score from 0.895 to 0.913, and the $F_\beta$ score from 0.787 to 0.833, and reduces the MAE score from 0.105 to 0.083.

| $\mathcal{C}$ | 2 | 4 | 6 | 8 | 10 |
|---|---|---|---|---|---|
| IoU ↑ | 0.629 | **0.632** | 0.631 | 0.631 | 0.628 |
| Accuracy ↑ | 0.903 | 0.905 | **0.908** | 0.908 | 0.906 |
| $F_\beta$ ↑ | 0.818 | 0.820 | **0.825** | 0.821 | 0.818 |
| MAE ↓ | 0.093 | 0.088 | **0.089** | 0.090 | 0.090 |

Table 2: Abl. of pattern map dimensions.

**Qualitative Comparisons.** Figure 3 visually compares mirror detection results produced by our network and state-of-the-art methods on different input video frames. For these small mirrors in the first three rows, our method can detect more details of these small mirrors. For big mirror objects at the fourth and fifth rows, all compared methods tend to neglect parts of mirror regions, while our method achieves a more complete result. Moreover, for these input video frames at the last two lines, compared methods tend to wrongly identify non-mirror objects as the mirror areas, while our method has a more accurate mirror detection result since our model can learn the mirror-related knowledge from the training dataset.

**Parameter Scale and Inference Efficiency Comparisons.** FTM-Net is an efficient method with fewer parameters and a faster inference speed. Figure 4 (a) presents the $F_\beta$, IoU scores, along with the corresponding parameter scale of our FTM-Net and VMD-Net. Specifically, our FTM-Net achieves an 0.833 $F_\beta$ and a 0.649 IoU score but it has only 57.41M parameters and 35.35M trainable parameters. Furthermore,

| $k$ | 3 | 5 | 7 | 9 | 11 |
|---|---|---|---|---|---|
| IoU ↑ | 0.631 | 0.635 | 0.642 | **0.649** | 0.645 |
| Accuracy ↑ | 0.908 | 0.909 | 0.910 | **0.913** | 0.912 |
| $F_\beta$ ↑ | 0.825 | 0.826 | 0.830 | **0.833** | 0.829 |
| MAE ↓ | 0.089 | 0.085 | 0.083 | **0.083** | 0.084 |

Table 3: Abl. of video frame number.

we compare our FTM-Net against state-of-the-art methods in terms of frames per second (FPS). Figure 4 (b) and (c) demonstrate the IoU score, the $F_\beta$ score, and the FPS of our network and five state-of-the-art methods. Apparently, we can find that our network has higher IoU, $F_\beta$, and FPS scores than all five compared methods. These results underline the effectiveness of our design for the video mirror detection task.

### 4.3 ABLATION STUDY

**The Dimension of Pattern Map.** We use the hyper-parameter $\mathcal{C}$ to control the dimension of the pattern map. We fix the number of video frames $k$ as 3 to discuss $\mathcal{C}$. As shown in Table 2, when the $\mathcal{C}$ is 6, our network achieves the best performance in terms of $F_\beta$ and IoU. Hence, we empirically set the dimension of pattern map $\mathcal{C}$ to be 6 for all our experiments.

| Index | PC-DOM | | DGA | | IoU↑ | Accuracy↑ | $F_\beta$↑ | MAE↓ |
|---|---|---|---|---|---|---|---|---|
| | DOM | PC | Point | Pattern | | | | |
| M1 | | | | | 0.552 | 0.866 | 0.758 | 0.114 |
| M2 | ✓ | | | | 0.592 | 0.897 | 0.804 | 0.102 |
| M3 | ✓ | ✓ | | | 0.617 | 0.905 | 0.815 | 0.094 |
| M4 | ✓ | ✓ | ✓ | | 0.625 | 0.905 | 0.827 | 0.089 |
| M5 | ✓ | ✓ | | ✓ | 0.627 | 0.907 | 0.829 | 0.085 |
| Ours | ✓ | ✓ | ✓ | ✓ | **0.649** | **0.913** | **0.833** | **0.083** |

Table 4: Ablation study on different components of our proposed method on the VMD-D dataset.

**The Number of Input Video Frames.** Then, we fix the dimension of pattern map $\mathcal{C}$ as 6 to evaluate the hyper-parameter $k$, which denotes the number of input video frames. Table 3 shows the results of $F_\beta$, IoU, and the FPS (frames per second) when $k$ is increased from 3 to 11 gradually. We can observe that when the video frame number $k = 9$, our method achieves the highest IoU and $F_\beta$. Hence, we confirm $k = 9$ as our default setting.

| MODULE | IoU↑ | Accuracy↑ | $F_\beta$↑ | MAE↓ |
|---|---|---|---|---|
| Single Frame | 0.617 | 0.905 | 0.815 | 0.094 |
| Pooling (Boureau et al., 2010) | 0.622 | 0.905 | 0.819 | 0.089 |
| Memory (Oh et al., 2019) | 0.619 | 0.903 | 0.822 | 0.086 |
| PATrans (Wu et al., 2023) | 0.625 | 0.907 | 0.822 | 0.086 |
| DGA (ours) | **0.649** | **0.913** | **0.833** | **0.083** |

Table 5: Abl. of temporal fusion modules.

**The Effectiveness of Major Modules in FTM-Net.** We conduct ablation studies to evaluate the effectiveness of two major components (i.e., PC-DOM and DGA) of our network. As shown in Table 4, M1 utilizes the original image-based SegFormer (Xie et al., 2021). However, M1 cannot effectively address the video mirror detection task, as it fails to capture the temporal information between different frames and does not consider depth information as a prior cue. Regarding M2, we incorporate DOM prediction based on M1 to offer depth as prior information, which achieves improvements on all four metrics compared to M1. Then, for M3, we integrate pattern-compensated DOM to further enhance the depth information. M3 achieves scores of 0.617, 0.905, 0.819, and 0.094 for IoU, Accuracy, $F_\beta$, and MAE, respectively. Moreover, we use our proposed DGA as the temporal fusion module and confirm the effectiveness of point-wise affinity (M4) and pattern-wise affinity (M5) separately. Finally, our FTM-Net, built upon the PC-DOM and DGA modules, achieves state-of-the-art results on all four metrics.

**The Temporal Fusion Modules.** To validate the effectiveness of our proposed DGA module compared to other temporal information fusion modules, we chose average pooling (Boureau et al., 2010), memory mechanisms (Oh et al., 2019), and PATrans (Wu et al., 2023) as comparison modules. As shown in Table 5, our DGA module achieves superior performance on all four metrics since it not only considers the relationships among different points but also accommodates visual patterns.

**The Effect of Different Backbones.** Our FTM-Net is a versatile approach that can adapt to various backbones. As shown in Table 6, we choose ResNet101 (He et al., 2016) (Res101), SwinTransformer-Small (Liu et al., 2021) (Swin-S), CaFormer-Medium (Yu et al., 2024) (CA-M), SegFormer-B3 (Xie et al., 2021) (Seg-B3), and SegFormer-B2 (Xie et al., 2021) (Seg-B2) to analyze their performance and the corresponding parameter scale. Seg-B2 achieves performance comparable to that of the other backbones but with fewer parameters, only 57.41M. Therefore, we choose Seg-B2 as default setting.

| BACKBONES | #Para | IoU↑ | Accuracy↑ | $F_\beta$↑ | MAE↓ |
|---|---|---|---|---|---|
| Res101 (He et al., 2016) | 69.28M | 0.643 | 0.912 | 0.821 | 0.088 |
| Swin-S (Liu et al., 2021) | 92.32M | 0.648 | 0.912 | 0.828 | 0.085 |
| CA-M (Yu et al., 2024) | 98.65M | 0.651 | **0.917** | **0.838** | 0.082 |
| Seg-B3 (Xie et al., 2021) | 73.01M | **0.654** | 0.915 | 0.833 | **0.079** |
| Seg-B2 (ours) | **57.41M** | 0.649 | 0.913 | 0.833 | 0.083 |

Table 6: Abl. of different backbones.

**The Label Efficiency of FTM-Net.** Our FTM-Net can achieve comparable video mirror detection performance with fewer labeled data, as it utilizes the pattern-compensated depth map as prior information. As shown in Figure 5, our FTM-Net outperforms other methods in terms of IoU score with only 40% labeled training data. Meanwhile, for the $F_\beta$ score, FTM-Net achieves a comparable performance with only 54% labeled training data.

**Visualization for Pattern-compensated Depth of Mirror.** Our proposed pattern-compensated decoder outputs multiple potential pattern maps to refine the original depth map. As shown in Figure

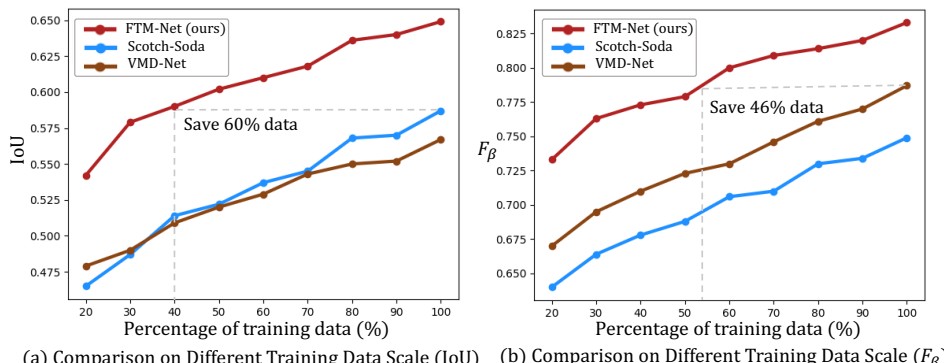

(a) Comparison on Different Training Data Scale (IoU)    (b) Comparison on Different Training Data Scale ($F_\beta$)

Figure 5: Visualization of data efficiency for our FTM-Net.

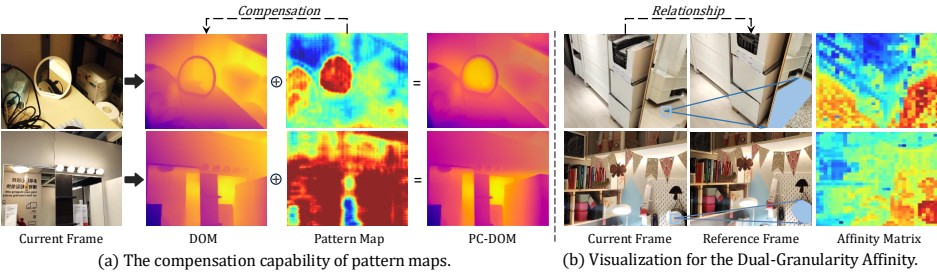

(a) The compensation capability of pattern maps.    (b) Visualization for the Dual-Granularity Affinity.

Figure 6: Visualization of the predictions made by the Pattern-Compensated decoder and the Dual-Granularity Affinity module.

6 (a), in some cases objects outside mirrors might have significant depth. Hence, the corresponding pattern maps can offer the compensation information for more accurate mirror localization.

**Visualization for Dual-granularity Affinity.** The dual-granularity affinity module can effectively model the relationship between different video frames. As illustrated in Figure 6 (b), the dual-granularity affinity module allows for the detection of similar objects between the current and reference frames. The corresponding active regions are prominent, showcasing the contextual information integration capability of our method.

## 5 BROADER IMPACTS AND LIMITATIONS

As a research work in video mirror detection, we believe this paper will not negatively impact society.

**Broader Impacts.** Our method can improve visual perception and safety for automated applications. Accurate mirror detection is crucial for applications such as drone tracking and robot navigation. It helps in avoiding potential safety issues by providing additional reference information.

**Limitations.** Like other methods, our method can detect mirrors well by introducing temporal DOM as guidance, but it might struggle with the complete details of the mirrors in some complex scenes.).

## 6 CONCLUSION

In this work, we introduce FTM-Net, a novel framework for effective mirror localization in videos. Inspired by a simple visual clue, we designed a Pattern-Compensated DOM estimation strategy to enhance mirror detection accuracy in single images and a Dual-Granularity Affinity module to track and integrate DOM changes across video frames. Empirical results demonstrate that FTM-Net significantly outperforms 18 leading state-of-the-art methods in video mirror detection.

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

## A  PYTORCH CODE

In this section, we will show the core Pytorch code of our Dual-Granularity Affinity module and the corresponding methodology analysis.

**Algorithm 1:** Point-wise Affinity

```python
def point(F_I_D, F_V):
  """F_I_D: (C, HW), F_V: (C, kHW)"""
  F_V_l2 = F_V.pow(2).sum(dim=0).unsquzzez(dim=1) # (kHW, 1)
  matrix = F_V.transpose().matmul(F_I_D) # (kHW, HW)
  S_Po = 2 * matrix - F_V_l2
  W_Po = torch.exp(S_Po) / torch.exp(S_Po.sum(dim=1))
  F_Po = F_V.matmul(W_Po) # (C, HW)
  return F_Po
```

**Algorithm 2:** Pattern-wise Affinity

```
def pattern(F_I_D, F_V):
  """F_I_D: (C, HW), F_V: (C, kHW)"""
  F_V_l2 = F_V.transpose().pow(2).sum(dim=0).unsquzzez(dim=1) # (C, 1)
  F_V_temporal = F_V.reshape(C, HW, k).mean(dim=-1) # (C, HW)
  matrix = F_V_temporal.matmul(F_I_D.transpose()) # (C, C)
  S_Pa = 2 * matrix - F_V_l2
  W_Pa = torch.exp(S_Pa) / torch.exp(S_Pa.sum(dim=1))
  F_Pa = F_V_temporal.transpose().matmul(W_Pa) # (C, HW)
  return F_Pa
```

Drawing inspiration from STCN Cheng et al. (2021b), we have simplified the computation of $S^{Po}$ and $S^{Pa}$ on the above code. The proof is shown as follows:

$$
\begin{aligned}
\mathbf{W}^{Po}_{i,j} &= \frac{\exp\left(\mathbf{S}^{Po}_{i,j}\right)}{\sum_n \exp\left(\mathbf{S}^{Po}_{n,j}\right)} \\
&= \frac{\exp\left(2(F^V)^T_i \cdot (F^{I+D}_t)_j - \left\|(F^V)^T_i\right\|^2_2 - \left\|(F^{I+D}_t)_j\right\|^2_2\right)}{\sum_n \exp\left(2(F^V)^T_n \cdot (F^{I+D}_t)_j - \left\|(F^V)^T_n\right\|^2_2 - \left\|(F^{I+D}_t)_j\right\|^2_2\right)} \\
&= \frac{\exp\left(2(F^V)^T_i \cdot (F^{I+D}_t)_j - \left\|(F^V)^T_i\right\|^2_2\right) / \exp\left(\left\|(F^{I+D}_t)_j\right\|^2_2\right)}{\sum_n \exp\left(2(F^V)^T_n \cdot (F^{I+D}_t)_j - \left\|(F^V)^T_n\right\|^2_2\right) / \exp\left(\left\|(F^{I+D}_t)_j\right\|^2_2\right)} \\
&= \frac{\exp\left(2(F^V)^T_i \cdot (F^{I+D}_t)_j - \left\|(F^V)^T_i\right\|^2_2\right)}{\sum_n \exp\left(2(F^V)^T_n \cdot (F^{I+D}_t)_j - \left\|(F^V)^T_n\right\|^2_2\right)}.
\end{aligned}
\tag{6}
$$

$$
\begin{aligned}
\mathbf{W}^{Pa}_{i,j} &= \frac{\exp\left(\mathbf{S}^{Pa}_{i,j}\right)}{\sum_n \exp\left(\mathbf{S}^{Pa}_{n,j}\right)} \\
&= \frac{\exp\left(2\psi(F^V)_i \cdot (F^{I+D}_t)^T_j - \left\|\psi(F^V)_i\right\|^2_2 - \left\|(F^{I+D}_t)^T_j\right\|^2_2\right)}{\sum_n \exp\left(2\psi(F^V)_n \cdot (F^{I+D}_t)^T_j - \left\|\psi(F^V)_n\right\|^2_2 - \left\|(F^{I+D}_t)^T_j\right\|^2_2\right)} \\
&= \frac{\exp\left(2\psi(F^V)_i \cdot (F^{I+D}_t)^T_j - \left\|\psi(F^V)_i\right\|^2_2\right) / \exp\left(\left\|(F^{I+D}_t)^T_j\right\|^2_2\right)}{\sum_n \exp\left(2\psi(F^V)_n \cdot (F^{I+D}_t)^T_j - \left\|\psi(F^V)_n\right\|^2_2\right) / \exp\left(\left\|(F^{I+D}_t)^T_j\right\|^2_2\right)} \\
&= \frac{\exp\left(2\psi(F^V)_i \cdot (F^{I+D}_t)^T_j - \left\|\psi(F^V)_i\right\|^2_2\right)}{\sum_n \exp\left(2\psi(F^V)_n \cdot (F^{I+D}_t)^T_j - \left\|\psi(F^V)_n\right\|^2_2\right)}.
\end{aligned}
\tag{7}
$$

## B FUTURE WORK

In the future, we plan to explore the data scaling law in the context of video mirror detection tasks for zero-shot scenarios and further enhance the aggregation of temporal information by considering longer sequences of frames within a video. Lastly, we aim to develop an algorithm that is more efficient for a variety of applications.

## C MORE VISUAL COMPARISONS

This section provides more visual comparisons between our FTM-Net and the only video mirror detection method: VMD-Net. As shown in Figure 9, 10, 8, our method can accurately detect

the location of mirrors through the pattern-compensated DOM feature. In addition, our FTM-Net can segment more complete mirror areas since it considers the temporal changes of the pattern-compensated DOM from both point and pattern perspectives, facilitated by the proposed Dual-Granularity Affinity module.

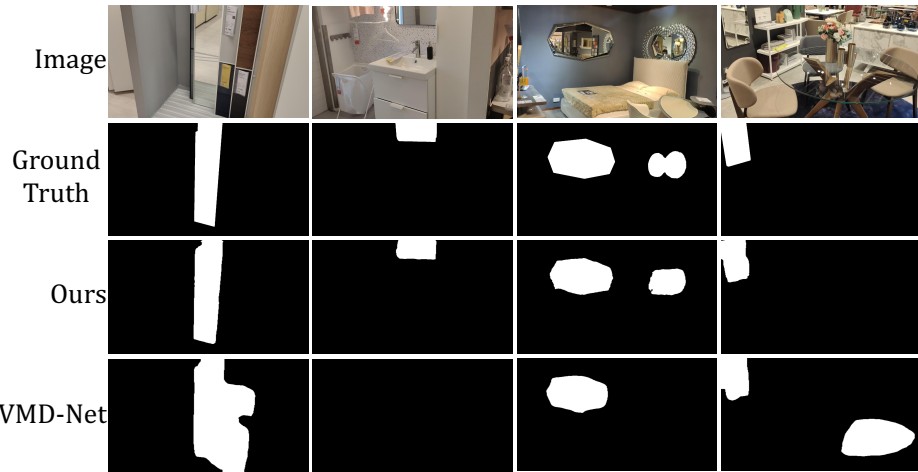

Figure 7: More visual comparisons against the VMD-Net.

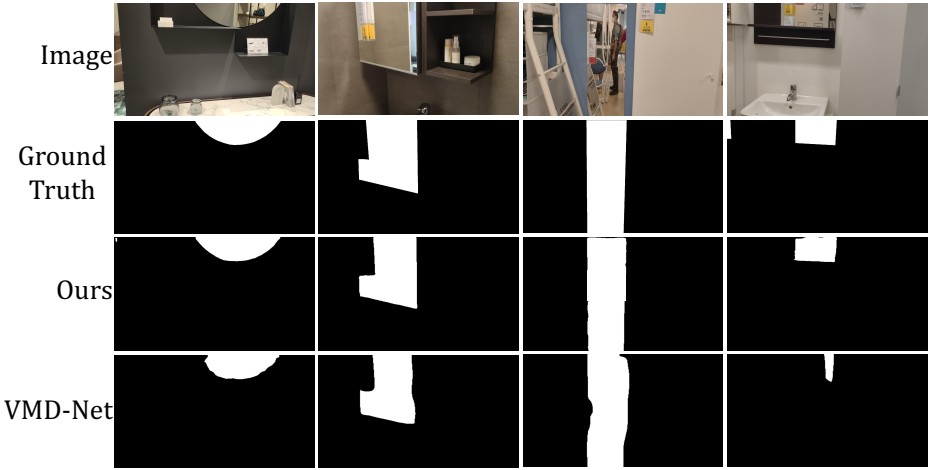

Figure 8: More visual comparisons against the VMD-Net.

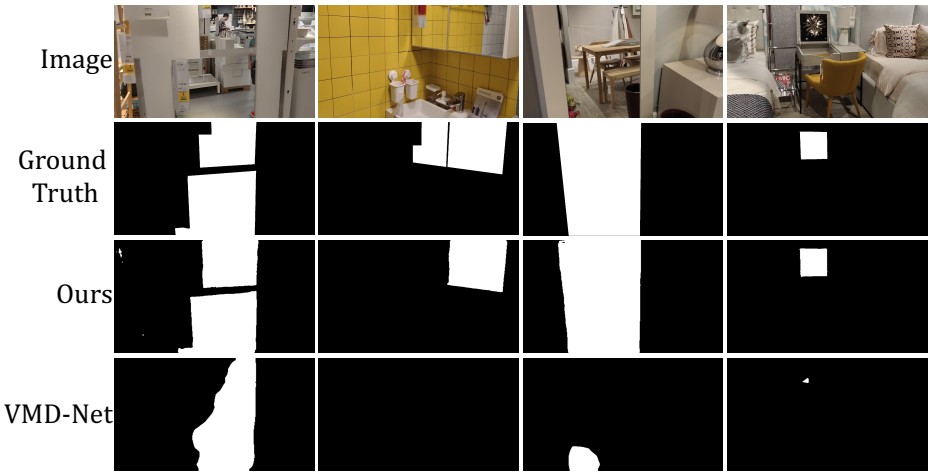

Figure 9: More visual comparisons against the VMD-Net.

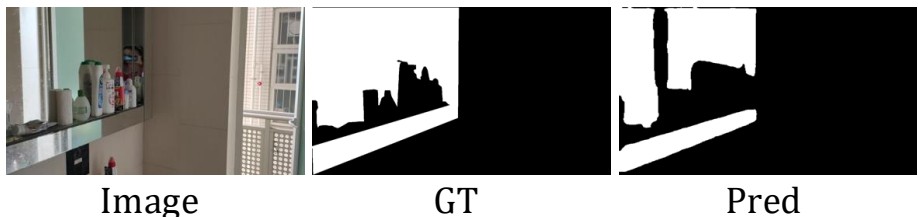

Figure 10: Failure cases in some complex scenes.

