# OpenReview forum: "Farther Than Mirror: Explore Pattern-Compensated Depth of Mirror with Temporal Changes for Video Mirror Detection"
_ICLR.cc/2025/Conference — ICLR 2025 Conference Withdrawn Submission_

### Official Review · Reviewer_bTZh · 2024-10-19

**Soundness:** 3
**Presentation:** 3
**Contribution:** 2
**Rating:** 5
**Confidence:** 5

**Summary:**

This paper presents FTM-Net, a novel video mirror detection model that improves mirror localization accuracy through Depth of Mirror (DOM) analysis. Key contributions include a Pattern-Compensated DOM estimation strategy for refining DOM in single images and a Dual-Granularity Affinity module for detecting mirrors in video sequences by tracking DOM changes. Experiments show that FTM-Net outperforms 18 state-of-the-art methods in video mirror detection.

**Strengths:**

1. The writing is clear and easy to understand.
2. A wide range of comparison methods are selected, demonstrating the superiority of the proposed method.
3. Introducing large vision models into the Mirror Detection field is insightful and offers new directions.

**Weaknesses:**

1. In fact, the integration of depth signals into Mirror Detection (MD) has been explored in earlier work [1], so the innovation based on depth difference in this paper is limited.
2. Recent work on Video MD, such as [2], has not been included, nor has validation been performed on that dataset.
3. The selection of comparison methods is somewhat outdated; for example, the latest work in IMD includes CSFwinformer [3] and DPRNet [4], and the most recent SOD methods are not from 2020 alone.
4. The authors should note that in the VMDNet work, the input image size is 384×384, whereas in this paper, it is set to 512×512. This is unfair.

In summary, this paper is conventional and has room for improvement.

[1]. Depth-Aware Mirror Segmentation, CVPR2021

[2]. Effective Video Mirror Detection with Inconsistent Motion Cues, CVPR2024

[3]. CSFwinformer: Cross-Space-Frequency Window Transformer for Mirror Detection, TIP2024

[4]. Dual Domain Perception and Progressive Refinement for Mirror Detection, TCSVT2024

**Questions:**

Please see the weaknesses mentioned above, and additionally, the authors are advised to carefully review the latest advancements in the related field.

---

### Official Review · Reviewer_jTGr · 2024-10-19

**Soundness:** 3
**Presentation:** 3
**Contribution:** 2
**Rating:** 3
**Confidence:** 5

**Summary:**

This paper proposes a method (called FTM-Net) for detecting mirror regions in videos based on a simple observation that the perceived depth in mirror regions are often larger than mirror boundaries. Therefore, FTM-Net first uses DepthAnything (Yang et al., 2024) to estimate the depths of video frames, which are combined with RGB features for mirror detection. The FTM-Net refines the DepthAnything-predicted depth by predicting mirror-related features. The paper extends the point-wise temporal affinity computation (originally proposed for video object segmentation by (Cheng et al., 2021b)) by computing pattern-wise affinity.

While this paper handles a relatively new task, ***I am leaning to reject this paper due to (1) insignificant main idea due to unreliable observation and unclear discussion w.r.t. (Mei et al., 2021); (2) limited technical novelty; and (3) unconvincing results due to missing important comparisons and some comparisons are unfair.***

**Strengths:**

+This paper handles a relatively new but challenging task that detects mirror regions from RGB videos.

+The proposed method is relatively easy to understand, and the codes are provided.

**Weaknesses:**

***Insignificant Main Idea.***
The paper is based on a simple observation that “the contents in mirrors tend to have larger depth values than mirrors”, and then introduce depth estimation for mirror detection.

However, this observation may not help differentiate mirror regions with other objects such as an open door/window or a glass window. It seems that the proposed method cannot handle such scenes.

Incorporating depth for mirror detection has been explored in (Mei et al., 2021), despite that the method of (Mei et al., 2021) uses depth as input and handles only single input image while the proposed method handles videos and uses depth estimator. The observation made in (Mei et al., 2021) that “ToF depth estimates do not report the true depth of the mirror surface but instead return the total length of the reflected light paths” is the same to this work. The paper does not differentiate itself from (Mai et al., 2021) by simply saying their work requires depth input.

***Limited Technology Novelty.***
This paper claims two novelties, the Pattern-Compensated DOM estimation and the Dual-Granularity Affinity module. However, the Pattern-Compensated DOM estimation uses pre-trained DepthAnything (Yang et al., 2024) to predict depths, and duplicates the decoder of DepthAnything to predict mirror patterns for refining depths. The Dual-Granularity Affinity module extends the point-wise affinity to pattern-wise by adding a temporal pooling. These designs are rather minor.

***Unconvincing Results.***
First, although 18 existing methods are used for comparisons, the results of using baseline model (i.e., segFormer (Xie et al., 2021), M1 in Table 4) have already outperformed most of these competing methods reported in Table 1.
Second, while those competing methods are re-trained on the VMD-D dataset, which only contains RGB information, the proposed method uses additional pre-trained depth estimator. The proposed method is not compared to RGB-depth based methods, for example, (Mei et al., 2021) with depth predicted from DepthAnything as input.

***Minor issues.***
The pattern map dimensions and the number of input video frames seem not affect the detection results much according to the Table 2 and 3. Hence, I suggest to move these two Tables to the supplemental. This could save a lot of space for discuss important results.

There is a “).” at the end of the last sentence in Section 5.

**Questions:**

***Questions/suggestions.***
1. Please clarify whether the proposed method could differentiate mirrors from open/glass doors and windows with examples and analysis.
2. Please explain the specific technical novelties and add a comparison to (Mei et al., 2021).
3. Please explicitly explain why the PC DOM estimation and DGA module are novel.
4. Please clarify why the baseline model outperforms many existing methods and discuss the implications of this.
5. Please add comparisons to RGB-depth-based methods like Mei et al. (2021), using DepthAnything for depth estimation to ensure a fair comparison.
6. It is not clear how the mirror patterns are learned via the pattern decoder. Are there any supervision?
7. It is not clear how the mirror patterns are used to refine DOM map into Pattern-Compensated DOM after concatenation.
8. According to Figure 2 there should be multiple mirror pattern maps but there is only one mirror pattern map per scene in Figure 6. Any reason why?
9. It is better to visualize affinity maps using only point-wise, pattern-wise, and both.
10. The paper misses three recent mirror detection works:

[A] Effective Video Mirror Detection with Inconsistent Motion Cues, CVPR 2024

[B] ZOOM: Learning Video Mirror Detection With Extremely-Weak Supervision, AAAI 2024

[C] Self-supervised Pre-training for Mirror Detection, ICCV 2023

---

### Official Review · Reviewer_3wYQ · 2024-10-27

**Soundness:** 2
**Presentation:** 2
**Contribution:** 2
**Rating:** 5
**Confidence:** 4

**Summary:**

This article suggests that objects reflected in a mirror seem to be positioned farther away than the mirror itself. Building on this idea, the authors investigate the mirror's depth to improve its localization and use pattern maps to adjust for depth, enhancing mirror location accuracy. Additionally, the article introduces a dual-granularity affinity module that integrates both pixel and pattern variations of video features into the current frame feature. Both modules perform effectively, and the overall approach achieves state-of-the-art results in video mirror detection.

**Strengths:**

1. Introducing the depth of mirrors into semantic segmentation, and comprehensively discussing the possible relationship between the depth of mirrors in images and the depth of other objects, to assist in locating mirrors and better achieve mirror detection.
2. By fusing features in both pixel level and pattern level between frames, the state of the art results of video mirror detection has been achieved.

**Weaknesses:**

1. The channel dimension of PC-DOM obtained by simply concatenating pattern maps and DOM is unclear, and the visualization is a single channel depth map, but in reality it should be multi-channel.
2. Pattern maps rely on the pattern decoder, and more space is needed to explain how end-to-end supervised training ensures the reliability of pattern maps.
3. Pattern maps rely on the features of depth estimation models, while Depth Anything has the ability to recognize partial mirrors and correctly predict mirror depth. Whether the extracted features are beneficial for mirror detection needs more discussion.
4. The prerequisite for prior applicability is that there is a mirror in the input image. If there is no mirror, more explanation is needed to determine whether the model will mistakenly recognize distant objects as mirrors.

**Questions:**

1. Could you please explain the channel dimension of PC-DOM in detail?
2. Could you please offer more explanation about the reliability of Pattern maps? Better in visualization.
3. When the DOM is correct which is the depth of the mirror itself, how the PC-DOM and the overall method will perform?
4. When there is no mirror, will the model predict a mirror in mistake? Because the model is based on the prior that the depth of mirror is often to be farther away.

---

### Note · Authors · 2024-11-13

I have read and agree with the venue's withdrawal policy on behalf of myself and my co-authors.